# Investigating the causal effect of maternal continuum of care on child's minimum acceptable diet: A multilevel approach using partially pooled propensity score weighting

**Shafayet Khan Shafee**[1]*, **Md. Niamul Islam Sium**[2], **Bishal Sarker**[1], **Riyadul Islam**[3]

**1** Institute of Statistical Research and Training, University of Dhaka, Dhaka, Bangladesh, **2** International Centre for Diarrhoeal Disease Research Bangladesh, Dhaka, Bangladesh, **3** Uttara University, Dhaka, Bangladesh

* sshafee@isrt.ac.bd

## Abstract

Malnutrition contributes to half of child mortality in low- and middle-income countries like Bangladesh. It's challenging for a developing country to improve child nutrition using limited resources and other difficulties. This underscores the importance of developing and implementing targeted interventions that effectively address these constraints. This study addresses this gap by investigating whether a mother receiving complete continuum of care comprising antenatal care, skilled birth attendance, and postnatal care improves the child's likelihood of achieving a minimum acceptable diet, an indicator combining minimum dietary diversity and minimum meal frequency. Analyzing data of 6,494 mother-child pairs from the 2019 Bangladesh Multiple Indicator Cluster Survey, we applied a multilevel modelling approach with partially pooled propensity score weighting to control for potential confounders and account for district-level variations. The results show that mothers who received complete continuum of care increased their children's chances of meeting minimum acceptable diet requirements by 17% (ATE [95% empirical bootstrap CI]: 1.17 [1.01, 1.34]). This study is among the first to explore the causal link between mother's receiving complete continuum of care and minimum acceptable diet intake of children using multilevel data. The findings should support policymakers in making informed decisions to improve child nutrition by ensuring comprehensive maternal care. Sensitivity analysis ensures that the observed effect estimate is robust to unmeasured confounding.

## 1 Introduction

Childhood undernutrition is a serious public health issue, particularly in developing nations [1]. Globally, it contributes to nearly half of all deaths among children under five [2]. Two-thirds of these deaths are linked to poor feeding practices and

**Data availability statement:** The data used in this study to support the findings are available at the following Figshare link https://doi.org/10.6084/m9.figshare.30073489.

**Funding:** The author(s) received no specific funding for this work.

**Competing interests:** The authors have declared that no competing interests exist.

associated infectious diseases [3]. The World Health Organization (WHO) and the United Nations Children's Fund (UNICEF) highlight the first 1,000 days of life as a crucial period for nutritional interventions. During this time, the brain undergoes its most rapid growth, and malnutrition can result in stunting and hindered developmental outcomes [4]. The WHO guidelines for Infant and Young Child Feeding (IYCF) recommend exclusive breastfeeding for the first six months of life. After six months, breastfeeding should continue alongside other nutritional sources until at least the age of two [5].

The Minimum Acceptable Diet (MAD) is an indicator used by the WHO to assess child feeding practices. It combines both Minimum Dietary Diversity (MDD) and Minimum Meal Frequency (MMF) [6]. Dietary diversity reflects both the quality and quantity of a diet, as well as food security, and serves as a valuable indicator of micronutrient adequacy [7]. MDD was originally defined as the consumption of at least four out of seven food groups to ensure adequate daily nutrition and higher dietary quality. The seven food groups included grains, roots and tubers; legumes and nuts; dairy products (e.g., infant formula, milk, cheese, yogurt); flesh foods (meat, fish, poultry, and organ meats); eggs; vitamin A–rich fruits and vegetables; and other fruits and vegetables [6]. In 2017, the WHO Technical Expert Advisory Group on Nutrition Monitoring (TEAM) revised this definition to incorporate breast milk as an additional food group. Accordingly, MDD is now defined as the consumption of at least five out of eight food groups by a child [8].

MMF is an indicator of a child's energy source which examines the number of times a child received other foods than breast milk [9]. The minimum meal is specific to the age of the child and breastfeeding status. Breastfed children aged 6–8 months meet the MMF criterion if they consume solid, semi-solid, or soft foods at least twice a day, while those aged 6–23 months must consume such foods at least three times a day. For non-breastfed children aged 6–23 months, the MMF criterion is met if they receive solid, semi-solid or soft foods or milk feeds at least four times a day where at least one of the feeds must be a solid, semi-solid, or soft feed [10]. In this context, milk feeds refer to any formula (e.g., infant formula, follow-on formula, toddler milk) or any animal milk other than human breast milk (e.g., cow's milk, goat's milk, evaporated or reconstituted powdered milk), as well as yogurt and other fermented milk products in semi-solid or drinkable form [10].

MAD incorporates both MDD and MMF to evaluate a child's nutritional adequacy. For breastfed children, MAD is achieved when both the MDD and MMF requirements are satisfied. For non-breastfed children, MAD is met when they fulfill the MDD and MMF requirements and additionally consume at least two milk feeds [10]. MAD is well established indicator for evaluating child feeding practices presented via the WHO. It is one of the eight core indicators of complementary feeding outlined in WHO's guidelines for infant and young child feeding (IYCF) practices for children aged 6–23 months [6]. Children who fulfill MAD requirements are less likely to be underweight [11].

A continuum of care (CoC) approach to maternal health is being advocated as a crucial program strategy to ensure that women receive essential services throughout pregnancy, childbirth, and the postpartum period [12]. The healthcare services

a woman receives throughout the continuum of maternal care during pregnancy, childbirth, and the immediate postnatal period are vital for the survival and well-being of both the mother and the child [13]. CoC is achieved when a mother receives at least four antenatal care (ANC) visits (at least one from skilled provider), is attended by a skilled birth attendant during delivery, and receives postnatal care (PNC) within 48 hours [14]. CoC is considered incomplete if any of these conditions are not met. For instance, if a mother misses ANC visits, does not receive PNC, or lacks skilled delivery care, the CoC is not fulfilled [15,16].

Several factors, including maternal education, wealth index, place of residence, maternal age, media exposure, and parity (number of children) significantly influence both the completion of the CoC and the likelihood of a child receiving the MAD [17,18]. Media exposure has a notable impact on both CoC completion and MAD attainment [19,20]. Additionally, both the unwanted pregnancies and parity influence the likelihood of CoC completion and meeting the MAD criteria [18–20]. Moreover, the education level of the household head may influence both CoC and MAD, as a more educated head is typically better equipped to manage family resources, thereby increasing the likelihood of the child receiving the necessary care and nutrition.

ANC visits have a significant effect on MAD [21]. Mothers who attended four ANC visits were twice as likely to provide MAD compared to those who had fewer than four ANC visits [22]. On the other hand, skilled birth attendance (SBA) impacts maternal health, and PNC visits also show a strong association with MAD [23,24]. Mothers who attended PNC visits were more likely (AOR 1.68) to provide MAD [25]. Since CoC consists of ANC visits, SBA and PNC visits, it is high likely that CoC is associated with MAD. It is crucial to emphasize that the existing literature has only explored associative relationships between the components of CoC (i.e. ANC visits, SBA, and PNC visits) and MAD. To the best of our knowledge, no research to date has measured the causal effect of CoC on MAD, which serves as the primary motivation for our study.

This study seeks to determine the causal effect of mothers' receiving complete CoC (exposure) on MAD intake of children (outcome) using observational multilevel data from a cross-sectional survey. Additionally, the presence of hierarchical structure of the data further motivates our research. To investigate the causal relationship from an observational multilevel data, it is essential to adopt a multilevel framework for causal inference that can account for potential confounders by effectively balancing them and can provide a controlled descriptive comparison of the outcome between exposed and unexposed group [26]. Confounding is a concern in all observational studies related to causality [27]. Various methods have been developed to adjust for confounding effects in observational studies. In this study, we apply inverse probability weighting (IPW) method to achieve covariate balance [26]. Furthermore, this study use partially pooled IPW method to adjust for unmeasured cluster level covariates [28].

Potential concerns may arise regarding the causal pathway between mothers' receiving CoC and children's MAD intake, particularly with respect to how the temporal ordering of exposure and outcome can be inferred from a cross-sectional survey. Previous studies have shown that components of CoC, such as antenatal care, skilled delivery, and postnatal care, are associated with improved child feeding practices [21–25]. In the Bangladeshi context, where maternal and child health services are closely linked, these associations provide a strong rationale for examining the potential causal effect of CoC on child nutritional outcomes. Although the cross-sectional design does not directly establish temporality, the CoC components necessarily occur during pregnancy, delivery, and the early postnatal period, whereas MAD is measured among children aged 6–23 months. This temporal sequence mitigates the risk of reverse causation and supports the plausibility of causal inference based on cross-sectional data.

## 2 Methods

### 2.1 Data source

The current study utilized observational secondary data from the Bangladesh Multiple Indicator Cluster Survey (MICS) 2019. This is a cross-sectional household survey conducted by the Bangladesh Bureau of Statistics (BBS) in

collaboration with UNICEF Bangladesh, as part of the Global MICS programme. Data were collected between January 19 and June 1, 2019, in a single round, with no repeated follow-up interviews [29]. A two-stage stratified cluster sampling design was employed, in which 3,220 primary sampling units (clusters) were selected, from which 64,400 households were enumerated nationwide. Since the sample is not self-weighting, the survey data provides sampling weights for valid population-level estimates. Standardized questionnaires were administered to women aged 15–49 years and caregivers of children under five, capturing information on maternal and child health, nutrition, and related socioeconomic indicators, including items from which CoC completion of mothers and MAD intake of children can be derived. The official survey findings report "Progotir Pathey, Bangladesh Multiple Indicator Cluster Survey 2019" provides a detailed account of the sampling procedure and data description [29]. To ensure respondent privacy, unique identifiers such as names and location details collected during interviews were removed from the survey data [29]. These anonymised data files are freely accessible from the MICS website (https://mics.unicef.org/surveys). The current analysis focused on mothers who had a live birth within the two years preceding the survey and their children aged 6–23 months, resulting in a complete sample of 6,494 observations extracted from the survey data.

**Ethical considerations and approval.** The survey protocol was approved by the BBS. In addition to obtaining verbal agreement from each participant, informed consent was secured from a parent or guardian for children aged below 18 years, prior to seeking their own consent to participate. All respondents were informed that their participation was voluntary and their responses would remain confidential. They were also free to skip any question or withdraw from the interview at any point during the survey. For the present study, no additional ethical approval was required because the analysis was based on publicly available, de-identified secondary data from the MICS 2019 survey.

## 2.2 Variables

The outcome variable for the analysis was MAD intake of a child, coded as 0 if the child did not meet MAD requirements and 1 if they did. Children aged 6–23 months were classified as receiving MAD if they met criteria for both MDD and MMF. Completion of maternal CoC was considered as the exposure variable. This was coded as 1 when a mother received all three essential maternal health care, including four or more ANC visits (at least one from skilled provider), childbirth attended by a skilled birth attendant, and a postnatal checkup within the 48 hours after delivery. If any of these services were missing, the CoC was considered incomplete and coded as 0. Skilled providers in this context refer to health professionals such as medical doctors, nurses, midwives, paramedics/medical assistants, family welfare visitors, and community-based skilled birth attendants [29]. A skilled birth attendant refers to a doctor, nurse, or midwife with accredited training who is competent to manage normal pregnancies, childbirth, and the immediate postnatal period, and who is also qualified to recognize, mangae, and refer complications affecting mothers and newborns [30].

Upon reviewing the literature [17–20], the following variables were identified as observed confounders, each assumed to be associated with both exposure and outcome: place of residence (urban or rural); mother's exposure to media (yes or no), including newspapers, magazines, radio, or TV; mother's age; number of children ever born (1, 2, 3+); mother's wealth index category (poor, middle, rich); whether the mother wanted the last child (yes or no); education level of the household head (< secondary or ≥ secondary); and mother's education level (< secondary or ≥ secondary).

## 2.3 Propensity score estimation with clustered data

The current analysis employed an inverse probability weighting (IPW) approach [31] to estimate the causal impact of mothers' receiving complete CoC on children's MAD intake. Given that the data has a hierarchical structure where the children and their mothers can be considered nested within the districts, a multilevel regression modeling approach is essential for accurate estimation of the propensity scores [26]. Furthermore, as the data was derived from a complex survey design that includes sampling weights [29], these weights should be incorporated into the propensity score estimation stage to ensure representative analysis [32]. Therefore, we used a weighted multilevel regression model to

estimate the propensity scores, integrating the sampling weights as model weights [33] and considering the districts as the clusters.

## 2.4 Partially pooled propensity scores

Given the two-level structure of the data, where mothers and children are nested within the districts, we index observations using dual subscripts: $h$ for districts ($h = 1, 2, \ldots, H$, where $H$ represents the total number of districts) and $k$ for individuals within each district ($k = 1, 2, \ldots, n_h$, where $n_h$ denotes the sample size for district $h$). Let $n = \sum_h n_h$ denote the total sample size. Define $A_{hk}$ as a binary variable indicating whether the $k^{th}$ mother in district $h$ received complete CoC (treatment group, where $A = 1$) or did not receive complete CoC (control group, where $A = 0$). Define $P_h := \sum_{k=1}^{n_h} A_{hk}/n_h$ as the treatment prevalence within each district $h$. The binary outcome variable $Y_{hk}$ is observed for each child. Let $\mathbf{X}_{hk}$ represent the vector of observed individual-level covariates, with each element $x_{hk}$ corresponding to a specific covariate value for the $k^{th}$ mother-child pair in district $h$. Due to data limitations, this analysis could not directly incorporate any cluster-level covariates (i.e., covariates whose values are the same for all observations nested within each district). Let $\mathbf{U}_h$ represent these unmeasured cluster-level covariates.

To account for confounding bias due to unmeasured cluster-level covariates ($\mathbf{U}_h$), we applied a partially pooled propensity score estimation method [28]. This approach selectively pools clusters into groups based on the clusters' treatment prevalence, $P_h$, and then fits a propensity score model for each group, incorporating all observed covariates and random effects to estimate group-specific propensity scores. This approach reduces bias from unmeasured cluster-level confounding and improves the precision of propensity score estimates by ensuring adequate sample size within each group, compared to cluster-specific propensity score estimation [28].

We applied the $k$-means clustering algorithm [34] to group the districts (i.e., clusters) based on their treatment prevalence, $P_h$. Let $H_g$ represent the set of districts within group $g$, where $g = 1, \ldots, G$ and $G \leq H$. To determine the number of groups, $G$, we used the Elbow method [35] in conjunction with visual inspections. For each group, we then fitted a weighted multilevel regression model incorporating all observed individual-level covariates (e.g. $\mathbf{X}_{hk}$) to estimate the group-specific propensity scores. Let $e_{hk(\text{group})}$ denote the propensity score estimated for $k^{th}$ mother-child pair in district $h$ for the group to which district $h$ was assigned.

## 2.5 Estimating causal effect using IPW

Our target estimand, $\pi$, is the average treatment effect (ATE) of mothers' receiving complete CoC on children's MAD intake in the population. We considered the following group-weighted estimator [28] for ATE,

$$\hat{\pi}_{\text{group}} := \frac{\sum_{g=1}^{G} \hat{W}_{g(\text{group})} \hat{\pi}_{g(\text{group})}}{\sum_{g=1}^{G} \hat{W}_{g(\text{group})}}, \tag{1}$$

where $\hat{\pi}_{g(\text{group})}$ is the estimated ATE for group $g$,

$$\hat{\pi}_{g(\text{group})} := \frac{\dfrac{\sum_{h \in H_g} \sum_{k=1}^{n_h} A_{hk} \hat{w}_{hk(\text{group})} Y_{hk}}{\sum_{h \in H_g} \sum_{k=1}^{n_h} A_{hk} \hat{w}_{hk(\text{group})}}}{\dfrac{\sum_{h \in H_g} \sum_{k=1}^{n_h} (1 - A_{hk}) \hat{w}_{hk(\text{group})} Y_{hk}}{\sum_{h \in H_g} \sum_{k=1}^{n_h} (1 - A_{hk}) \hat{w}_{hk(\text{group})}}} \tag{2}$$

and the group weight $\hat{w}_{g(\text{group})}$ is the sum of the weights of all the mother-child pairs in the group $g$, $\hat{w}_{g(\text{group})}$ $:= \sum_{h \in H_g} \sum_{k=1}^{n_h} \hat{w}_{hk(\text{group})} = \sum_{h \in H_g} \hat{w}_{h(\text{group})}$. The inverse probability weight for each mother-child pair in group $g$ is defined as follows [28]:

$$\hat{w}_{hk(\text{group})} := \frac{A_{hk}}{\hat{e}_{hk(\text{group})}} + \frac{1 - A_{hk}}{1 - \hat{e}_{hk(\text{group})}} \tag{3}$$

Using the group-weighted IPW estimator ($\hat{\pi}_{\text{group}}$) allows us to mitigate the influence of unmeasured district-level covariates ($\mathbf{U}_h$) on causal effect estimation, particularly when $\mathbf{U}_h$ modifies causal effects. This is achieved by first estimating treatment effects locally within each group of clusters sharing similar $P_h$ values and then marginalizing over these groups to obtain the ATE [28].

## 2.6 Sensitivity analysis

The presence of unmeasured confounding variables can substantially compromise the validity and reliability of causal effect estimates in observational research. To address this issue, Kasza, Wolfe and Schuster developed an approach (the KWS framework) to evaluate the influence of unmeasured confounding in scenarios involving binary exposure and outcome [36].

The KWS framework utilizes sensitivity analysis parameters to determine reasonable bounds for causal effects [36]. However, the causal effect in the original population may not correspond to that in the weighted population, potentially leading to a misrepresentation of the original population when applying this approach. To address these limitations, adjustments to the KWS framework have been proposed for evaluating unmeasured confounding in binary outcome settings [37]. In our research, we applied this modified framework to conduct a sensitivity analysis of unmeasured confounding. Based on this approach, we first determined the range of the sensitivity parameter $K_a$, as outlined below:

$$\max_{\mathbf{x}} \hat{P}r(A = a \mid \mathbf{X}) \leq K_a \leq \min_{\mathbf{x}} \left( \hat{P}r(A = a \mid \mathbf{X}) + \frac{\hat{P}r(A = 1 - a \mid \mathbf{X})}{\hat{P}r(Y = 1 \mid A = a, \mathbf{X})} \right),$$

for $a \in \{0, 1\}$ [37]. Using this range, we proceeded to determine the ranges for the average counterfactual outcomes: $E(Y^{a=1})$ and $E(Y^{a=0})$, and consequently assessed the sensitivity of our observed results to unmeasured confounding.

## 3 Analysis and results

### 3.1 Creating groups

To estimate the partially pooled propensity scores, we first calculate the MAD intake prevalence for each district. Districts are then grouped based on their mean MAD intake prevalence using the $k$-means algorithm. The decision regarding the number of groups is guided by visual inspections of the elbow plot and the bar plot (Figs 1 and 2, respectively).

Based on the within-group homogeneity and between-group heterogeneity observed in these figures, either 4 or 5 groups could be considered. However, to ensure a larger number of districts within each group (indicated by the labels at the top of each bar in Fig 2), we selected 4 groups for our analysis.

### 3.2 Estimating propensity scores and checking covariate balance

After forming the groups, we fit a weighted two-level random intercept logistic regression model within each group to estimate the propensity scores ($\hat{e}_{hk(\text{group})}$). The inverse probability weights ($\hat{w}_{hk(\text{group})}$) for each mother-child pair are then calculated using Eq (3). These weights are used to evaluate covariate balance within each group. Fig 3 presents love plots for each group, showing both unweighted and weighted absolute mean differences across covariates.

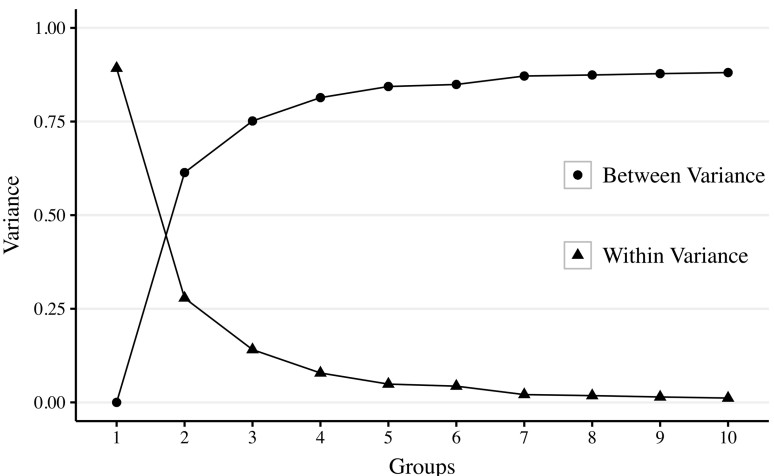

**Fig 1**. **Variance patterns for *k*-means clustering.** Filled circles represent between-group variance, and filled triangles represent within-group variance, across different numbers of *k*-means groups.

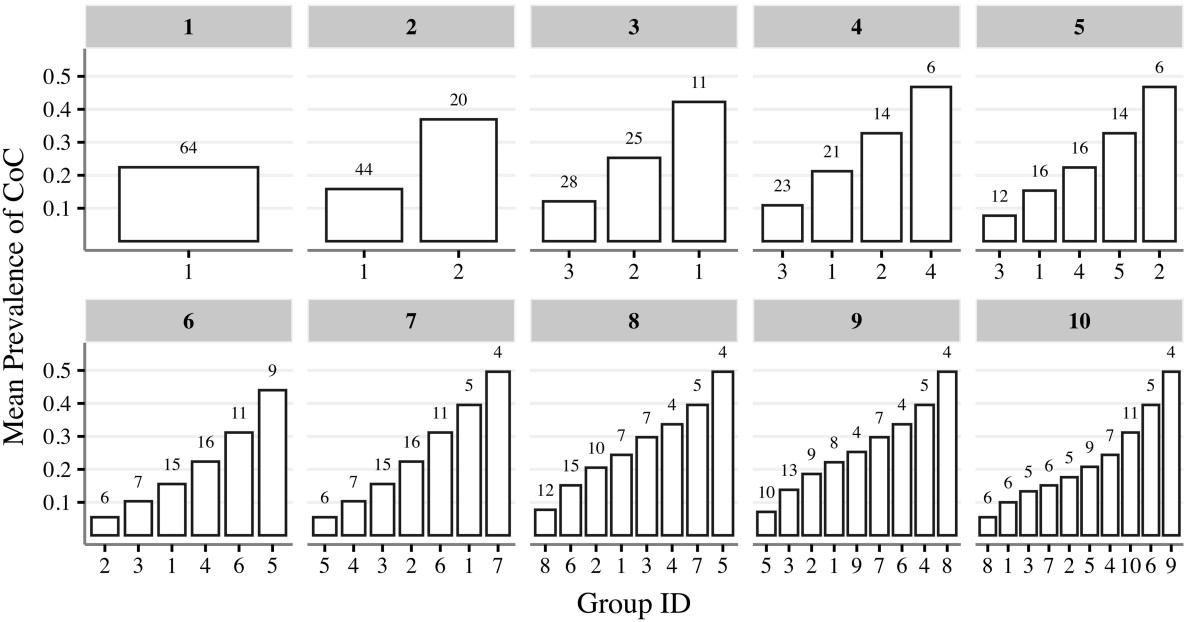

**Fig 2**. **Mean prevalence of CoC across groups created using the *k*-means algorithm.** Each figure panel corresponds to a specific number of *k*-means centroids considered when forming the groups, as indicated by the label at the top of each panel. The x-axis denotes the group ID assigned by the *k*-means algorithm, and the numbers at the top of each bar indicate the number of districts (i.e., clusters) within each group

The weighted absolute mean differences are below the threshold of 0.1, indicating covariate balance [38–40]. This is further corroborated by the mirror histograms of weighted propensity scores for each group in Fig 4, which illustrates the similarity in the distribution of weighted propensity scores between the mothers who did and did not receive the complete CoC.

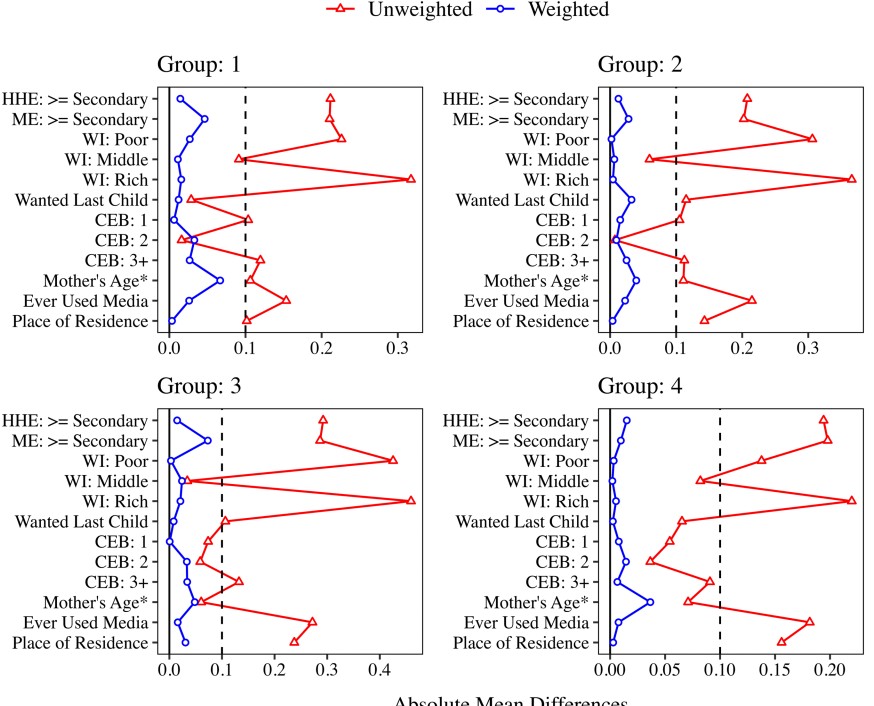

**Fig 3**. **Absolute mean differences in covariates for mothers who received and did not receive complete CoC.** Results are shown for unweighted and weighted samples across four groups (Groups 1 to 4). The vertical dashed line represents an absolute mean difference of 0.1. HHE refers to household head's education level, ME to mother's education level, WI to wealth index, and CEB to the number of children ever born. Since "Mother's Age" is a continuous variable, its absolute mean difference is standardized and marked with an asterisk (*).

### 3.3 Estimating causal effect

Using the calculated inverse probability weight for each mother-child pair within each group, we obtain group weight ($\hat{w}_{g(\text{group})}$) and group specific estimate of ATE ($\hat{\pi}_{g(\text{group})}$) as defined in Eq (2). Then following Eq (1), we calculate the group-weighted estimate of ATE, representing the causal effect of mothers receiving complete CoC on children's MAD intake, to be 1.17. This result suggests that a child is 17% more likely to achieve MAD if their mother received complete CoC compared to a child whose mother did not receive the complete CoC. To calculate the confidence interval for this group-weighted estimate of ATE, we performed 2,000 bootstrap iterations. The distribution of the 2,000 bootstrap-generated ATE estimates is shown in Fig 5. The 95% empirical bootstrap confidence interval for the ATE is (1.01, 1.34). Since this interval does not include one, the estimated effect can be considered statistically significant.

We also performed a sensitivity analysis to assess whether our observed effect estimates remained robust or were influenced by unmeasured confounding, applying the framework described in [37]. The analysis yielded the following ranges for the sensitivity parameters: $K_0 \in [0.98, 1.18]$ and $K_1 \in [0.85, 1.06]$. The global lower and upper bounds of the causal effect on the risk ratio scale were determined to be: [1.16,1.74]. Our findings suggest that the causal effect of CoC on MAD is robust to unmeasured confounding, as the estimated ranges do not include the null value of 1.

## 4 Discussion and conclusion

### 4.1 Discussion

This study examined the causal impact of mothers' completion of the CoC on the likelihood of their children meeting the MAD requirements. Ensuring adequate child nutrition, especially through practices like the MAD, is still a major

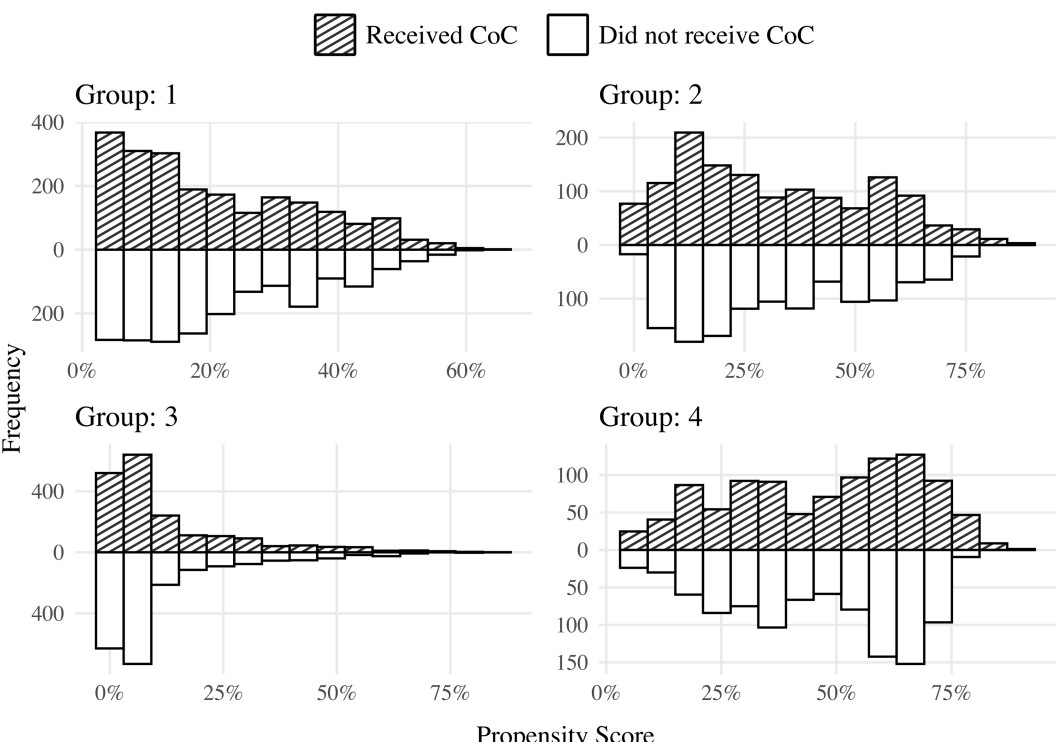

**Fig 4**. **Mirror histograms of propensity scores (weighted by inverse probability weight) for mothers who did and did not receive the complete CoC.** Each panel corresponds to a specific group (1 to 4), displaying the distribution of propensity scores within each group.

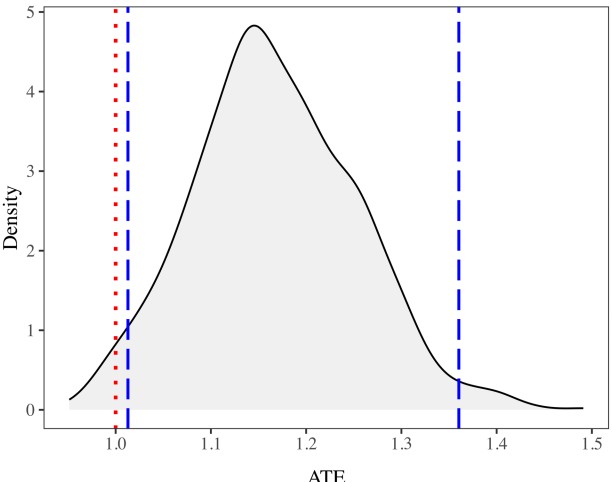

**Fig 5**. **Density plot of ATE estimates from 2,000 bootstrap iterations.** The vertical dotted line represents an ATE value of 1, which denotes no causal effect, while the dashed lines indicate the 2.5$^{th}$ and 97.5$^{th}$ percentile values, respectively, marking the empirical 95% confidence interval.

challenge in many developing countries, including Bangladesh. Limited resources and low awareness mean that many caregivers find it difficult to meet the nutritional standards essential for their children's growth and development. This gap highlights the pressing need for policymakers to find strategies that are not only effective but also practical and affordable

for widespread adoption. The CoC approach is one such strategy, as it brings together key elements that support child health. Because CoC is both low-cost and requires minimal resources, it has the potential to be a sustainable and scalable solution. For this reason, CoC is examined as a primary exposure in this study to explore how it might improve MAD and, by extension, child nutrition outcomes.

A partially pooled propensity score model was employed in this study to address the hierarchical structure of the data and account for cluster-level unobserved confounders, yielding robust estimates of ATE. A key strength of this approach is its ability to manage cluster-level variations and reduce potential biases from small sample sizes within individual clusters by pooling information across similar clusters. The results indicated a positive and significant relationship between mothers' completion of the CoC and the likelihood of their children meeting the MAD. Specifically, the children of mothers who completed CoC had a 17% higher likelihood of meeting MAD compared to those whose mothers did not. Furthermore, the stability of these findings was confirmed through bootstrap procedures, reinforcing the validity of the observed treatment effect.

The findings align with existing literature that associates improved maternal healthcare practices, such ANC visits, SBA, and PNC, with better child health outcomes [21–25]. Our findings suggest that comprehensive maternal care plays a critical role in ensuring that children receive appropriate nutrition from complementary meals alongside breastmilk during the first two years of life, a crucial period for child development. This emphasizes the importance of promoting maternal healthcare interventions to improve child nutrition outcomes, particularly in low-resource settings.

Strengthening achievement of the CoC in Bangladesh requires policy actions that address both health system capacity and the broader social determinants of maternal care. Expanding access to ANC in rural and underserved areas can be facilitated through mobile clinics, which have been shown to be essential and well-accepted services for women in remote communities [41], and by strengthening the role of community health workers (CHWs), whose home visits and collaboration with clinical providers have been shown to improve ANC uptake and maternal health outcomes [42]. Improving maternal referral systems between local clinics and hospitals with skilled birth attendants is also critical for ensuring safer deliveries and continuity of postnatal care. Addressing barriers such as weak communication, transportation challenges, and limited healthcare system capacity is central to enhancing referral effectiveness [43]. In parallel, CHWs should be empowered with clearly defined scopes of practice and tailored training, as their trusted position within communities makes them well suited to deliver maternal and newborn health services, provided that responsibilities remain aligned with their training duration and national regulations [44].

Beyond service delivery, broader social and economic measures are equally important for improving CoC completion. Women's empowerment, including education, access to information, and social independence, has consistently been associated with adherence to maternal care across pregnancy, delivery, and postpartum stages [45]. Targeted health promotion programs for women with low education levels or limited media exposure, alongside policies to raise the age at first marriage, can further strengthen uptake of services [45]. Economic interventions, such as microcredit, livelihood support, conditional cash transfers, and voucher programs, have also demonstrated effectiveness in enabling women to access and complete the full CoC [46,47]. Finally, developing an integrated home-to-hospital healthcare system and offering adequate incentives for health professionals to work in rural communities can strengthen service continuity, reduce gaps in care, and promote equitable access [45,48]. Collectively, these measures could substantially enhance CoC completion and, in turn, improve maternal and child health outcomes in resource-limited settings.

### 4.2 Limitations

However, certain limitations must be considered. First, although the temporal ordering between the exposure and the outcome in this study is established by their definitions, effect estimates derived from cross-sectional survey data may still be subject to recall bias [49,50], as maternal care practices and child feeding were self-reported by respondents. Secondly, the MAD serves as a proxy measure to determine if children had a diverse diet and met the minimum meal frequency in

the 24 hours prior to the survey. However, it only captures dietary information just before the survey date. Additionally, the relatively small sample sizes in some clusters may affect the precision of the estimates, despite the adjustments made by the partially pooled model. Lastly, we did not account for any cluster-level covariates. The number of clinics or health complexes, for instance, is an important cluster-level factor. However, due to a lack of such data in the MICS dataset, we were unable to include it in our analysis. Future studies could address this limitation by incorporating longitudinal data or using larger datasets to verify the long-term impact of CoC on child nutrition.

### 4.3 Conclusion

This study demonstrates the significant positive impact of completing the CoC on child nutrition outcomes, particularly in ensuring that children meet the MAD requirements. By highlighting the critical role of comprehensive maternal healthcare services, our findings emphasize the need to strengthen policies and programs that promote access to and completion of the full spectrum of maternal care. Such efforts have the potential to yield lasting improvements in child health and development in Bangladesh and other resource-limited settings. Future research should first explore broader causal pathways linking maternal care to child nutrition and developmental outcomes, and should also explore advanced causal inference methods, such as matching, overlap weighting, and machine learning approaches for refining propensity score estimation, to strengthen and consolidate the evidence base for maternal and child healthcare interventions.

### Author contributions

**Conceptualization:** Shafayet Khan Shafee, Md. Niamul Islam Sium.

**Data curation:** Shafayet Khan Shafee.

**Formal analysis:** Shafayet Khan Shafee.

**Methodology:** Shafayet Khan Shafee, Md. Niamul Islam Sium.

**Software:** Shafayet Khan Shafee.

**Visualization:** Shafayet Khan Shafee.

**Writing – original draft:** Shafayet Khan Shafee, Md. Niamul Islam Sium, Bishal Sarker, Riyadul Islam.

**Writing – review & editing:** Shafayet Khan Shafee, Md. Niamul Islam Sium, Bishal Sarker, Riyadul Islam.

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
