## [Decision Letter · Decision Letter 0]

27 Aug 2025

PONE-D-25-28053Investigating the Causal Effect of Maternal Continuum of Care on Child's Minimum Acceptable Diet: A Multilevel Approach using Partially Pooled Propensity Score WeightingPLOS ONE

Dear Dr. Shafee,     

Thank you for submitting your manuscript to PLOS ONE. After careful consideration, we feel that it has merit but does not fully meet PLOS ONE’s publication criteria as it currently stands. Therefore, we invite you to submit a revised version of the manuscript that addresses the points raised during the review process.

Could you provide details on how the survey data were collected, for example, was this a once off survey? Were the repeated rounds of data collection etc.?. I couldn't figure this out in your methods section as it currently stands. I also struggled to see how you infer "causality" on what seems to me was a cross-sectional survey, hence the comment for you to provide details on the design of the "survey"/study. At the end of this letter, you will find reviewer comments. Please make sure to address all the comments as they are valid and complimentary between the two reviewers. 

We look forward to receiving your revised manuscript.

Kind regards,

Hanani Tabana, Ph.D

Academic Editor

PLOS ONE

Journal Requirements:

Additional Editor Comments:

Could you provide details on how the survey data were collected, for example, was this a once off survey? Were the repeated rounds of data collection etc.?. I couldn't figure this out in your methods section as it currently stands. I also struggled to see how you infer "causality" on what seems to me was a cross-sectional survey, hence the comment for you to provide details on the design of the "survey"/study.

Reviewers' comments:

Reviewer's Responses to Questions

**Comments to the Author**

1. Is the manuscript technically sound, and do the data support the conclusions?

Reviewer #1: Yes

Reviewer #2: Yes

2. Has the statistical analysis been performed appropriately and rigorously? 

Reviewer #1: I Don't Know

Reviewer #2: Yes

3. Have the authors made all data underlying the findings in their manuscript fully available?

Reviewer #1: Yes

Reviewer #2: Yes

4. Is the manuscript presented in an intelligible fashion and written in standard English?

Reviewer #1: Yes

Reviewer #2: Yes

5. Review Comments to the Author

Reviewer #1: Well done on your manuscript.

I am not well versed on the statistical analysis used in the study hence I said, 'I don't know' to the question 'Has the statistical analysis been performed appropriately and rigorously?'

Introduction

1. Specify whether the associations you mention here are positive or negative, don't leave the reader guessing for e.g. is the influence of social media, parity, maternal age etc. on CoC and MAD a positive or negative one?

Variables

1. What is a skilled provider or skilled birth attendant

2. What are milk foods?

Sensitivity analysis

1. Second sentence -'the approach introduced by?' Please revise this sentence.

Methods

1. MICS data collection was done from Jan to June 2019 - It is important to acknowledge the presence of Covid 19 during data collection. How does this affect what is being observed by the study?

Consistency of terms

coc and mad vs CoC and MAD - rather stick to one way of writing these terms so that you do not lose the reader.

Reviewer #2: Thank you for the opportunity to review this manuscript. The authors provide an interesting methodology to evaluate the causal relationship between the maternal continuum of care and the child’s minimum acceptable diet. The article is well written, concepts are well explained, and comprehensive analysis has been done. The methods section contains a thorough explanation of the variables included, and confounders that have been accounted for, and a detailed description of the analysis approach taken. The use of secondary data for this analysis is commendable.

Below I have included some comments:

Introduction:

Pg 2, line 16-22: Please rephrase, to make it clear that MDD previously represented the consumption of four out of seven groups, but is now classified as the consumption of five out of eight food groups. Additionally, on pg 2, line 35-36 the definition “MDD requirement (consuming at least 4 out of 6 food groups, excluding dairy products)” seems to the explanation of MDD in the methods section (pg 4, line 120-124 where all 8 groups are included. Please clarify.

Pg 3 line 84: “Questions”, not “question” – meaning of sentence is unclear, please rephrase

Methods:

Ethical considerations: Although mention is made that the primary study had ethical clearance, it is unclear that permission was granted by study participants for this data to be used for the purpose of this secondary data analysis, please clarify.

There is repetition of how to classify MAD in the introduction, as well as the methods section. Although this is usually in the methods section, as other definitions are provided in the introduction, I suggest keeping the classification to the introduction to prevent repetition.

Figure 5: The figure heading states that the vertical dotted line represents an ATE value of 0, but it is plotted at 1.0 on the x-axis (which represents ATE according to the figure). Please clarify this.

Pg 8, Line 258: There is missing text, sentence ends with “framework introduced by”. Please rephrase.

Pg 8, Line 261: Please replace “coc on mad” with “CoC on MAD”

Discussion:

Given the findings of this study, and the resource-limited setting of Bangladesh, the discussion would be strengthened by a more robust examination of actions policymakers need to take to improve achievement of CoC.

Pg 8, line 290: I would argue that the statement “Our findings suggest that comprehensive maternal care plays a critical role in ensuring that children receive appropriate nutrition during the first two years of life”, should be expanded to include the first 1000 days (from conception to the age of two); given that CoC encompasses the antenatal period too.

6. PLOS authors have the option to publish the peer review history of their article (what does this mean?). If published, this will include your full peer review and any attached files.

Reviewer #1: No

Reviewer #2: No

---

## [Author Response · Author response to Decision Letter 1]

24 Sep 2025

Please note that we have uploaded a separate "Response to Reviewers" PDF file along with the revised manuscript. This file contains detailed responses to all comments and questions raised by the reviewers and the Academic Editor.

---

## [Editor Report · Decision Letter 1]

19 Oct 2025

Investigating the causal effect of maternal continuum of care on child's minimum acceptable diet: A multilevel approach using partially pooled propensity score weighting

PONE-D-25-28053R1

Dear Dr. Shafee,

We’re pleased to inform you that your manuscript has been judged scientifically suitable for publication and will be formally accepted for publication once it meets all outstanding technical requirements.

Kind regards,

Hanani Tabana, Ph.D

Academic Editor

PLOS ONE

---

## [Editor Report · Acceptance letter]

PONE-D-25-28053R1

PLOS ONE

Dear Dr. Shafee,

I'm pleased to inform you that your manuscript has been deemed suitable for publication in PLOS ONE. Congratulations! Your manuscript is now being handed over to our production team.

Kind regards,

on behalf of

Associate Professor Hanani Tabana

Academic Editor

PLOS ONE